# A Comparative Analysis of RNAi Trigger Uptake and Distribution in Mosquito Vectors of Disease

**DOI:** 10.3390/insects14060556

**Published:** 2023-06-15

**Authors:** Paul M. Airs, Katherine E. Kudrna, Bailey Lubinski, Yashdeep Phanse, Lyric C. Bartholomay

**Affiliations:** 1Department of Pathobiological Sciences, University of Wisconsin—Madison, Madison, WI 53706, USA; 2Midwest Center of Excellence for Vector-Borne Diseases, University of Wisconsin—Madison, Madison, WI 53706, USA

**Keywords:** RNA interference, dsRNA, biodistribution, per os, *Aedes*, *Anopheles*, *Culex*

## Abstract

**Simple Summary:**

RNA interference (RNAi) is a widely conserved antiviral mechanism whereby double-stranded RNA (dsRNA) is identified and degraded in cells through the RNAi pathway. Since the RNAi pathway uses dsRNA sequences to degrade a target, researchers can introduce dsRNA sequences that match genes of interest to study the impact of a gene. In mosquitoes, researchers introduce dsRNA to study an array of genes at different life stages and in different species, but the destination of dsRNAs in mosquitoes has yet to be characterized. In this work, dsRNA sequences were fluorescently labeled and tracked in mosquito species of medical significance, including *Aedes aegypti*, *Anopheles gambiae*, and *Culex pipiens.* Different exposure routes were trialed, including injection (peritoneal), feeding (*per os*), and topical application. Following injection, dsRNAs accumulate in a subset of cells associated with phagocytosis functions such as hemocytes and ovaries, but were not internalized following *per os* or topical routes. Additionally, Northern blotting was used to monitor the degradation and clearance of dsRNA following exposure in *Ae. aegypti*, identifying dsRNA in tissues up to a week post exposure, but were cleared from most individual tissues more rapidly. Overall, these findings describe the distribution of introduced dsRNAs in mosquitoes and may direct future research using this technique.

**Abstract:**

In mosquitoes, the utilization of RNAi for functional genetics is widespread, usually mediated through introduced double-stranded RNAs (dsRNAs) with sequence identity to a gene of interest. However, RNAi in mosquitoes is often hampered by inconsistencies in target gene knockdown between experimental setups. While the core RNAi pathway is known to function in most mosquito strains, the uptake and biodistribution of dsRNAs across different mosquito species and life stages have yet to be extensively explored as a source of variation in RNAi experiments. To better understand mosquito-RNAi dynamics, the biodistribution of a dsRNA to a heterologous gene, LacZ (iLacZ), was tracked following various routes of exposure in the larval and adult stages of *Aedes aegypti*, *Anopheles gambiae*, and *Culex pipiens*. iLacZ was largely limited to the gut lumen when exposed per os, or to the cuticle when topically applied, but spread through the hemocoel when injected. Uptake of dsRNA was noted in a subset of cells including: hemocytes, pericardial cells of the dorsal vessel, ovarian follicles, and ganglia of the ventral nerve cord. These cell types are all known to undergo phagocytosis, pinocytosis, or both, and as such may actively take up RNAi triggers. In *Ae. aegypti*, iLacZ was detected for up to one week post exposure by Northern blotting, but uptake and degradation drastically differed across tissues. The results presented here reveal that the uptake of RNAi triggers is distinct and specific to the cell type in vivo.

## 1. Introduction

RNAi is a highly conserved pathway that plays an essential role in invertebrate antiviral immunity [1,2,3,4,5,6]. In mosquitoes, RNAi is robust and plays a role in the clearance of arboviral infections, but can also function to maintain persistent viral infections [5,6,7,8]. The core RNAi pathway functions as a post-transcriptional gene silencing mechanism whereby long double-stranded RNA (dsRNA) sequences are processed into short-interfering RNAs (siRNA) by Dicer-2 [9], which are then used as a guide via Argonaute-2 and the RNA-induced silencing complex to identify and degrade matching mRNA transcripts, preventing their translation [10,11,12]. The presence of this pathway has facilitated reverse genetics research through the introduction of introduced RNAi triggers (typically dsRNA or siRNA) homologous to targeting gene sequences and resulting in the downregulation or ‘knockdown’ of target genes. Because any gene can theoretically be targeted in a sequence-specific manner, RNAi-based approaches for the control of pest and disease vector species, and pathogens therein, are in development [13,14]. 

While RNAi has become integral to basic and applied entomology, research is often hampered by variation in knockdown success and experimental outcomes between experiments and the species tested [13,14,15,16,17]. Knockdown of mosquito genes following exposure to RNAi triggers has been reported in a variety of species, strains, life stages, and tissues [13]. However, it is not known whether RNAi is mediated by direct uptake of the RNAi trigger by endocytosis, as suggested by Saleh et al., 2006, or by systemic RNAi requiring amplification and spread from hemocytes, as witnessed by Tassetto et al., 2018 [18,19]. In mosquitoes, RNAi knockdown success is linked to the route of exposure and the delivery system employed, with oral delivery systems, the most amenable to vector control applications [13,14], possibly limited by the expression of dsRNA-degrading nucleases in the midgut [20]. Additionally, differences in tissue responses have been noted, such as the salivary glands of *An. gambiae*, which are recalcitrant to siRNA uptake compared to the ovaries [21]. In *Ae. aegypti*, knockdown of the oxysterol binding protein and apolipoprotein genes was more effective in the abdomen than the head, midgut, and ovary [22]. These studies demonstrate the differences in RNAi efficacy at the tissue level, but may be impacted by tissue-specific target gene expression. How tissue differences relate to the distribution and integrity of RNAi triggers, and whether these differences are species specific, have yet to be extensively explored.

Despite the frequent implementation of RNAi triggers in gene knockdown assays, the uptake and spread of exogenously produced RNAi triggers in mosquitoes have not been well defined. In nematode species such as *Caenorhabditis elegans*, systemic RNAi responses are facilitated through RNAi trigger amplification and spread by RNA-dependent RNA polymerases (RDRP) and SID-1 dsRNA-gated channels [23,24]. Insects largely lack RDRPs [25], although recent evidence of eukaryotic RDRPs in the Diptera *Clunio marinus* and *Rhagoletis zephyria* indicates potential for unidentified RDRPs in other insect species [26]. Much more is known about insect RNAi responses to viral infections. In Diptera, a systemic response is vital to prevent viral superinfection, mediated by the amplification and spread of the viral dsRNA signal [2,8,27]. In this context, viral RNA can be taken up by circulating hemocytes, which launch a response through the production of viral RNAi triggers that are packaged into exosomes and passed to naive cells [19]. Understanding RNAi responses to viral infection does not explain the fate of exogenously produced non-viral RNAi triggers used in RNAi knockdown studies, especially when the spread of RNAi triggers is not facilitated by viral infection and responses. However, the passage of RNAi triggers via exosomes also occurs in other insects [28,29] as well as filarial nematodes [30,31], and therefore, there may be a conserved means of spreading RNAi triggers within and between species independent of virus infections.

We reasoned that delineating and describing RNAi uptake and efficacy in a comparative manner, across species and tissue types, would support more informed RNAi experimental design, as well as outline the basis for systemic RNAi in the absence of virus infection. To address this, we describe the distribution of heterologous LacZ RNAi triggers (iLacZ) in *Aedes aegypti*, *Anopheles gambiae*, and *Culex pipiens* tissues following intrathoracic injection, per os, or by topical exposure using a combination of fluorescence microscopy and Northern blot analyses.

## 2. Materials and Methods

### 2.1. Mosquito Rearing and Maintenance

*Aedes aegypti* (Liverpool strain), *Anopheles gambiae* (G3 strain), and *Culex pipiens* (Iowa strain) larvae were reared in enamel pans and fed daily with a slurry of ground TetraMin^TM^ (Blacksburg, VA, USA). Unless otherwise stated, groups of 50 female pupae were collected ~24 h prior to emergence and adults were maintained in cartons on a 10% sucrose diet. For blood feeding, individuals were starved for ~24 h and then exposed to defibrinated sheep blood (HemoStat Laboratories, CA, USA) maintained at 37 °C through a Parafilm M^®^ (Bemis, WI, USA) membrane, using a blown glass membrane feeder and circulating water bath. All life stages were maintained at 28 °C at 70% relative humidity with a 16:8 h (light:dark) photoperiod.

### 2.2. Cell Culture

*Aedes albopictus* C6/h36 cells were maintained in Leibovitz L-15 media (Corning, NY, USA) with 10% FBS kept in a humidified incubator at 28 °C. For dsRNA uptake and degradation, ~50,000 live cells, as estimated by hemocytometer readings with methylene blue staining, were seeded in triplicate per group in a 96-well plate and exposed to 1 µg 417 bp iLacZ at 0.25, 1, 24, and 72 h prior to harvesting. Cells were gently washed three times with PBS before RNA extraction in TRIzol™ (Invitrogen, Thermo Fisher Scientific, Waltham, MA, USA) with a chloroform isopropanol clean-up. Purified RNA was eluted in nuclease-free water and assessed by 1.2% TAE gel electrophoresis and Nanodrop 1000 spectrophotometry.

### 2.3. RNAi Trigger Synthesis and Labeling

LacZ RNAi triggers were produced using T7-tagged primers (Appendix A) targeting the LacZ region of the pGEM^®^-T Easy vector (Promega, Madison, WI, USA). LacZ PCR products were amplified using GoTaq^®^ Flexi polymerase (Promega, Madison, WI, USA), purified using the Wizard^®^ SV PCR Clean-Up kit (Promega, Madison, WI, USA), and were then subjected to the MEGAscript™ RNAi kit (Invitrogen, Thermo Fisher Scientific, Waltham, MA, USA) with a phenol:chloroform and isopropanol clean-up. For microscopy experiments, Cy3 fluorescent labels were conjugated to dsRNA (~10 fluorophores per 100 bp) using the Cy3 Label IT^®^ kit (Mirus, Madison, WI, USA) with ethanol precipitation. This approach has previously been shown to label dsRNAs without impairing knockdown in insects [32]. All PCR products and dsRNAs were resuspended in nuclease-free water and subjected to 1.2% TAE gel electrophoresis and quantification using a NanoDrop 1000 spectrophotometer (Thermo Fisher Scientific, Waltham, MA, USA) for quality control.

The degradation of iLacZ dsRNA in insectary conditions was also tested; 1 mg/mL of unlabeled iLacZ was incubated for 12 h in insectary conditions in either a closed micro/centrifuge tube, an open micro/centrifuge tube lid, or an open micro/centrifuge tube lid in a carton of 20 adult female *Ae. aegypti* that had been starved for 24 h prior. Samples were collected, subject to Nanodrop 1000 spectrophotometry, and run on a 1.2% TAE gel electrophoresis. A one-way ANOVA was performed to assess differences in band intensity following gel electrophoresis compared to frozen iLacZ stock.

### 2.4. RNAi Trigger Exposure of Adult-Stage Mosquitoes

Mosquitoes were intrathoracically injected via the cervical membrane with 800–2500 ng RNAi trigger (0.5 µL for *Ae. aegypti* and *Cx. pipiens*, 0.2 µL for *An. gambiae*) using pulled borosilicate glass capillary needles (Kwik-Fill™, World Precision Instruments, Sarasota, FL, USA) and a micromanipulator. Individuals were cold-anesthetized and held in a glass Petri dish on ice prior to injection with a Whatman filter to prevent condensation damage to mosquitoes. For peritoneal exposure, groups of 25 adult females were treated 3–5 days post eclosion. For bloodfeeding analyses, fully engorged females were selected and injected ~24 h post bloodfeeding.

Per os exposures were performed as previously described [33] with groups of 20 adult females starved for 1 day post eclosion and then provided with 25–50 µL of RNAi trigger (0.8 mg/mL) in 0.5 M sucrose via borosilicate glass capillary tubes for 24 h.

Topical exposure included groups of 50 adult females (3 days post eclosion) that were cold anesthetized on ice and then exposed to 0.5 µL of acetone:RNAi trigger mixtures (3:1 with 1 mg/mL RNAi trigger) directly placed on the dorsal thorax and abdomen. The mixture was allowed to dry before the mosquitoes were returned to cartons.

Following exposure to dsRNA, mosquitoes were maintained in cartons with a 10% sucrose and monitored daily for survival until the tissues were processed. Dead individuals were removed from the study. For all experimental groups, three biological replicates were performed using separate cohorts of individuals.

### 2.5. RNAi Trigger Exposure of Larval-Stage Mosquitoes

For peritoneal exposure, groups of 25 L4-stage larvae were intrathoracically injected via the cervical membrane in a minimal volume of water on a glass slide with a ~500 ng RNAi trigger in ~0.1 µL using pulled borosilicate glass capillary needles (Kwik-Fill™, World Precision Instruments, Sarasota, FL, USA) and a micromanipulator.

For the per os exposure of larvae, a soaking method was adapted from a previous report [34]. Freshly hatched larvae were provided with a small amount of yeast extract in double distilled water. Then, 100 µL volumes of water containing 10 first instar/neonate *Ae. aegypti* larvae were transferred to a nuclease-free 1.5 mL microcentrifuge tube containing 10 µg of labeled iLacZ. Tubes were sealed and stored at the incubator settings described above.

### 2.6. Imaging of Live and Fixed Mosquito Tissues

At each timepoint, groups of 10 individuals were selected at random and dissected for analyses. Adult mosquitoes exposed to Cy3 conjugated iLacZ were cold-anesthetized at 4 °C and kept on ice prior to dissection. Individuals were held by a probe inserted through the thorax and dissected under a Zeiss Stemi 508 microscope. The head, legs, and wings were removed first using No. 5 forceps. Then, the seventh abdominal segment was cut and pulled to extract internal organs, including the alimentary tract, ovaries, and fat body, directly into room temperature nuclease-free PBS on a glass slide. The thoracic muscle and the abdomen were then bisected and sliced to reveal internal structures, and placed into PBS on a glass slide for observation. To determine auto-fluorescence and background noise, untreated control groups were used for *Ae. aegypti*, *An. gambiae*, and *Cx. pipiens* (see Appendix A). For larval analyses, groups of five were selected per timepoint. Larvae were transferred in water and dissected with micron pins by decapitation followed by cutting the seventh abdominal segment and the removal of internal structures. Soaked neonate larvae were imaged live in water as tissues were clearly visible through the cuticle. Samples were collected or observed from a minimum of five individuals per group per replicate. Dissected tissues were washed once in PBS, transferred to 70% ethanol cleaned glass slides in PBS, and covered with a coverslip. Slides were immediately imaged on a Zeiss Axio Scope.A1 with a QIClick™ CCD Camera (Q-imaging) and Nikon Elements D software. Image processing and representative panels were prepared using FIJI (version 1.54d 30 March 2023).

Fixed ovarian tissues from adult females were preserved (4% paraformaldehyde in PBS for 20 min), washed (3× in PBS), permeabilized (0.3% Triton X-100, 1% BSA, 1% Sodium citrate in PBS for 30 min and washed 3× in PBS), and then mounted on slides in ProLong™ Gold Antifade with DAPI (Invitrogen, Thermo Fisher Scientific, Waltham, MA, USA) prior to imaging on a Leica SP8 3X STED Confocal/Super-Resolution Microscope (Leica, Wetzlar, Germany).

### 2.7. Northern Blot Analyses

All surfaces and instruments were cleaned with ELIMINase^®^ (VWR, Radnor, PA, USA) prior to use. For Northern blot analyses, five individual samples were collected per group (timepoint or tissue type) from adult-stage mosquitoes and transferred to nuclease-free 1.5 mL tubes. For whole-body analyses, live cold anesthetized mosquitoes were directly homogenized in TRIzol™ by micropestles with a cordless microtube homogenizer (Bel-Art F65000-0000), taking care not to splash any material, and then centrifuged and another 400 μL TRIzol™ was added. For mosquito tissue analyses, extractions were performed as described above and immediately frozen on dry ice to minimize degradation during tissue extraction collection. Pooled tissues were thawed on ice in the presence of 800 μL TRIzol™ for extraction. For both tissues and whole mosquito lysates, samples were centrifuged and incubated for 5–10 min at RT to allow cell lysis to occur. RNA was purified by chloroform and isopropanol clean-up, and resuspended in nuclease free water, with some samples subject to DNase I treatment (MAXIscript™ T7 transcription kit, Invitrogen, Thermo Fisher Scientific, Waltham, MA, USA).

RNA was quantified by Nanodrop 1000 and standardized to the lowest sample concentration with a minimum threshold of 100 ng/µL. Then, 1–10 µg of RNA, standardized per gel, was prepared in a sample running buffer at 80 °C, chilled on ice, and then run on 1.2% TBE agarose gel at 90 v with ethidium bromide for 30–60 min on ice. Gels were imaged (AlphaImager HP) to determine RNA integrity and ribosomal band intensity, then transferred to BrightStar™ membrane using the NorthernMax™ kit (Invitrogen, Thermo Fisher Scientific, Waltham, MA, USA). Membranes were immediately UV crosslinked and hybridized in ULTRAhyb™ buffer (Invitrogen, Thermo Fisher Scientific, Waltham, MA, USA) at 68 °C overnight. T7 LacZ PCR products served as templates for the synthesis of ssRNA probes containing 40% biotin-14-CTP (MAXIscript™ T7 transcription kit (Invitrogen, Thermo Fisher Scientific, Waltham, MA, USA)). Probes were added after 30 min of hybridization and washed according to NorthernMax instructions. Membranes were developed using the Biotin Chromogenic Detection kit (Thermo Fisher Scientific, Waltham, MA, USA), imaged (AlphaImager HP), and aligned with the corresponding gel image using ImageJ and Adobe Photoshop CC to visualize RNA loading controls, ladder, and Northern dsRNA detection in one image. Unless otherwise stated, Northern blot images are representative of three biological replicates.

## 3. Results

### 3.1. Biodistribution of dsRNA in Ae. aegypti, An. gambiae, and Cx. pipiens

To assess and compare the distribution of exogenously produced RNAi triggers, fluorescently labeled iLacZ was produced and tracked in mosquito adults and larvae by epifluorescence microscopy in live tissues following exposure by injection, per os, and topical applications (Table 1, Appendix A).

#### 3.1.1. Tracking dsRNA following Injection in Adults and Larvae

Exposure by injection of both larval and adult stage *Ae. aegypti*, *An. gambiae*, and *Cx. pipiens* was performed and resulted in limited detectable uptake in numerous tissues. Several cell types most often displayed iLacZ signal, including pericardial cells, hemocytes, ovarian follicles, and ganglia of the ventral nerve cord (Table 1, Appendix A). However, iLacZ signal in the ganglia of the ventral nerve cord was not universally observed because we only noted this in *Ae. aegypti*, but not *An. gambiae* or *Cx. pipiens* (Table 1, Appendix A). Hemocytes containing iLacZ were detected throughout the body cavity in association with the head, legs, thorax, and fat body as well as with tracheoles in the ovaries and Malpighian tubules. Pericardial cells, hemocytes, and ovarian follicles were frequently detected with iLacZ signal in *Ae. aegypti*, *An. gambiae*, and *Cx. pipiens*, indicating that the uptake of RNAi triggers in mosquitoes is conserved.

In adults, excretion of dsRNA was seen from 72–120 h post exposure as iLacZ signal condensed in the alimentary tract lumen, suggesting that RNAi triggers or their degradation products are excreted from the hemolymph (Table 1, Appendix A). Providing a bloodmeal did not dramatically alter the destination of injected iLacZ, but the excretion of iLacZ from the hemolymph was not detectable in this condition (Table 1, Appendix A). However, digested blood in the alimentary tract could obscure the detection of excreted iLacZ in this case.

Fourth instar larvae of *Ae. aegypti*, *An. gambiae*, and *Cx. pipiens* injected with fluorescent iLacZ showed uptake in the same cell types as seen in adults, including pericardial cells, hemocytes, and the lumen of the alimentary tract (Table 1, Appendix A). Similar uptake patterns associated across species and life stages highlight specific and developmentally conserved mechanisms of trafficking RNAi triggers when presented to the hemolymph.

#### 3.1.2. Ovaries Appear to Be a Primary Destination of dsRNA

Ovaries appear to be a primary destination of iLacZ with and without a bloodmeal in *Ae. aegypti*, *An. gambiae*, and *Cx. pipiens* (Table 1). Providing a bloodmeal led to iLacZ uptake in the primary follicle oocytes of *Ae. aegypti*, *An. gambiae*, and *Cx. pipiens* as early as 1 h post exposure, which remained present throughout oogenesis (Table 1, Appendix A). As such, bloodfeeding may direct RNAi triggers to develop ovarian follicles, minimizing excretion. In addition, bloodfeeding also stimulates an increase in hemocyte proliferation from the fat body in mosquitoes [35]. Hemocytes containing iLacZ signal were found with and without a bloodmeal (Table 1, Appendix A). In *Cx. pipiens*, iLacZ was also occasionally detected in larger, high-lipid-content fat body cells.

Within the ovary, iLacZ was found in hemocytes associated with tracheoles, as well as pre-vitellogenic and post-vitellogenic oocytes of healthy and atretic follicles (see Appendix A). In *Cx. pipiens*, a facultatively-autogenous species, a stronger iLacZ signal was noted in oocytes of post-vitellogenic follicles in non-bloodfed individuals (see Appendix A). Other small cells appeared to contain an iLacZ signal, but could not be discerned by compound fluorescence microscopy.

To determine the sub-follicle localization of iLacZ, confocal microscopy was performed 24 h post injection in non-bloodfed *Ae. aegypti* ovaries (Figure 1 and Appendix A). Mosquito follicles are comprised of an oocyte, a follicular epithelial layer that traffics hormones and nutrients to the developing oocyte, and trophic cells (also known as nurse cells), that provide RNA and protein via cytoplasmic bridges to the oocyte [36,37]. While dsRNA was detected in both follicular epithelia and cytoplasm of oocytes, little to no signal was found in the nurse cells (Figure 1). iLacZ was also detectable in the follicular epithelium and oocytes of primary and secondary follicles. A strong signal was evident in atretic follicles (compare Figure 1 and Appendix A). Conversely, a minimal signal was detected in nurse cells or cells of the ovarian sheath. These results indicate that dsRNA in the hemolymph is transported to oocytes via the follicular epithelium and may not pass through the nurse cells.

#### 3.1.3. Assessing dsRNA Uptake by Topical and Per Os Exposure Routes

Topical iLacZ was administered to the dorsal thorax of adults using acetone as a carrier. The topical application of iLacZ resulted in the deposition of iLacZ on the cuticle of *Ae. aegypti*, *An. gambiae*, and *Cx. pipiens*, but no penetration of the cuticle to internal structures was found by fluorescence microscopy (Table 1, Appendix A).

Per os exposure of iLacZ in sucrose meals resulted in uptake to the ventral diverticulum and alimentary tract lumen of *Ae. aegypti* and *Cx. pipiens* (Table 1, Appendix A). Performing the per os assays required a period of starvation and exposure to a limited volume of iLacZ-sucrose medium, which hindered attempts to study uptake in *An. gambiae* due to high mortality. Uptake beyond the gut lumen was detected in the pericardial cells of *Ae. aegypti* and *Cx. pipiens* on occasion (Table 1, Appendix A). In *Cx. pipiens*, hemocytes in the thorax, legs, and ovaries contained iLacZ (Table 1, Appendix A).

The exposure of larvae to RNAi triggers can also be achieved by soaking first instar or neonate larvae in small volumes of water; this may result in dsRNA uptake per os. First, instar *Ae. aegypti* larvae were continuously exposed to nuclease-free water containing fluorescently labeled iLacZ for 72 h (Table 1, Appendix A). At 24 h post-exposure, the iLacZ signal was detected in the head and alimentary tract lumen, but no tissues beyond the lumen contained any detectable signal.

To determine whether dsRNA was degraded during per os exposures, a solution of iLacZ in sucrose was exposed to *Ae. aegypti* in insectary conditions. Neither insectary conditions nor the presence of mosquitoes significantly altered iLacZ integrity as assessed by gel electrophoresis, indicating that dsRNA is stable until ingestion using this assay (Appendix A).

### 3.2. Assessing the Persistence and Integrity of dsRNA in Mosquito Tissues and Cells

Tracking iLacZ by fluorescence microscopy allowed us to localize signal, but did not reveal information about the integrity or degradation of the RNAi trigger. To elucidate the persistence of RNAi triggers, Northern blots were performed in addition to fluorescence microscopy following exposure of iLacZ to *Ae. aegypti* adult females and Dicer-2 deficient *Ae. albopictus* C6/36 cells.

To determine whether tissues differentially process iLacZ, a 377 bp form of iLacZ was injected into adult female *Ae. aegypti* (Figure 2A). At 4 h post exposure, the signal was associated with all tissues. By 24 h post exposure, both the midgut and the carcass of the abdomen cleared most of the iLacZ signal, indicating either degradation or that full-length iLacZ was processed into undetectable siRNAs by this time. iLacZ persisted in all other tissues tested, including the head, thorax, and ovaries. iLacZ dsRNA products appear marginally larger due to the slower movement of dsRNA compared to dsDNA in low-density gels [38]. The detection of iLacZ in each tissue, however, does not indicate cellular uptake into target tissues, since free iLacZ in the hemolymph, hemocytes containing iLacZ, or iLacZ bound to the outer membrane of the tissue may all contribute to signal detection.

In addition to tissue processing, the persistence of iLacZ dsRNAs over time was measured in *Ae. aegypti* whole-body extracts up to 1 week post injection, with the same 377 bp form of iLacZ (Figure 2B). The degradation of iLacZ appears as a smear, which is evident in whole-body extracts from as early as 15 min post injection. A smaller band ~300 bp in length was evident, as seen in ovarian tissues at 24 h post exposure (Figure 2A).

To assess whether the persistence of the near full-length iLacZ signal was the result of DNA forms of dsRNA or from DNA contamination, DNase treatments of whole-body RNA extracts were performed at one week post injection (Figure 2C). The presence of iLacZ post-DNase treatment means the signal is either partially degraded dsRNA or may be subject to reverse transcription and re-expression; a phenomenon known to occur with viral dsRNAs [19]. The increased signal intensity associated with the DNase-treated sample is likely due to all samples being standardized by nanodrop, with carryover DNA altering the overall RNA quantity in the DNase-negative sample.

To determine whether smaller bands would persist from other dsRNAs, 200 bp, 400 bp, and 600 bp iLacZ products were produced and injected into adult female *Ae. aegypti* (Appendix A). In each case, a similar smear was seen in whole-body extracts, but other secondary smaller bands did not appear to accumulate following injection (Figure 2D).

Dicer-2 may act in preventing degradation by other nucleases, a phenomenon known to occur with long viral genomic dsRNA [39]. To determine whether degradation is impacted by non-Dicer-2 dsRNA nucleases, iLacZ was tracked following exposure to Dicer-2-deficient *Ae. albopictus* C6/36 cells [40] (Appendix A). In this case, all iLacZ signals were rapidly lost following exposure to C6/36 cells.

## 4. Discussion

In this study, we characterize the biodistribution and persistence of exogenously introduced dsRNA using the heterologous LacZ sequence in *Ae. aegypti*, *An. gambiae*, and *Cx. pipiens* mosquito vectors of medical significance. 

Following the injection of iLacZ, hemocytes, pericardial cells, ganglia of the dorsal vessel, as well as oocytes and follicular epithelia of ovarian follicles, were all destinations for fluorescent dsRNA (see Table 1). Uptake of iLacZ into these cell types may be driven by phagocytosis or pinocytosis, since pericardial cells [41], hemocytes [42], nerve glia [43,44], and ovarian follicles [45,46] have been shown to uptake cargo by these pathways. In *Drosophila* cells, various dsRNA uptake mechanisms have been identified including scavenger receptor-mediated endocytosis [47], RNAi trigger length-dependent endocytosis [18], and phagocytosis [48], and so the accumulation of iLacZ in phagocytic or pinocytic cells is likely following direct exposure of dsRNA to the hemolymph.

Hemocytes containing iLacZ were found throughout the body and were frequently detected in association with the trachea of the Malpighian tubules and ovaries. In larvae and adult *An. gambiae*, thin trachea are a site for hemocyte binding, as these sites are key for the entry of pathogens into the hemocoel [49]. Overall, the location of hemocytes varied. This could be due to individual variation, the number of hemocytes present in the body, hemocyte circulation at the injection site, or the creation of wounds, which could stimulate the production of hemocytes. As such, the lack of detection of hemocytes in some tissues, such as the legs, is in some specimens most probably due to differing numbers of hemocytes in circulation. As the legs were the first tissues to be removed during dissection, the desiccation of hemolymphs in the legs could have also occurred due to the hydrophobic nature of cuticle and cuticular hairs, leading to a loss of the punctate signal.

In addition to hemocytes, mosquito ovaries contained a large amount of fluorescent signal in all three species tested. This finding was surprising considering that ovaries are considered recalcitrant to dsRNA uptake in some species [22,50]; however, ovarian-derived cells are known to yield and have an active RNAi response [51]. However, *An. gambiae* ovaries are known to be more responsive to RNAi compared to tissues such as the midgut and salivary glands [21]. Mosquito ovaries are composed of ovarioles, strings of follicles where the terminal-most follicle (the primary follicle) develops first [37]. Each follicle contains an oocyte, follicular epithelia, and nurse cells. The follicular epithelial layer dictates the shape of the oocyte, provides the vitelline envelope, and mediates the uptake of nutrients from the hemolymph [46,52,53]. Nurse cells provide mRNA and other cytoplasmic contents to the oocyte through cytoplasmic bridges [54,55]. Following exposure to iLacZ dsRNA, we found the follicular epithelium and the oocyte to contain dsRNA, while the nurse cells remained relatively devoid of signal in bloodfed and non-bloodfed mosquitoes. It is possible that dsRNA is detected and transported to follicles as a nutrient source for oogenesis [36,56], and some developing follicles were seen containing dsRNA in *Cx. pipiens* ovaries, which are facultatively autogenous (see Appendix A). The mechanism for this transport, as well as what function dsRNA plays in triggering ovarian development or ovarian RNAi, has yet to be fully explored. It is known that ovaries are susceptible to RNAi (previously reviewed [13]), and it is possible that RNAi in the oocyte is essential to preventing superinfections in virus-infected mosquitoes.

Compared to injection, the distribution of iLacZ is mostly limited following per os and topical exposure routes for both larvae and adult-stage mosquitoes, but adult *Cx. pipiens* showed uptake into various tissues on occasion following per os exposure, indicating that per os may be a feasible exposure route in this species (see Table 1). Induced RNAi knockdown by per os exposure of naked dsRNA has been previously demonstrated in adult *Ae. aegypti* [57,58], so it may be that experimental conditions used in this study were not optimal for uptake, or that internalized dsRNA was not visible in comparison to the stronger signal present in the gut lumen. Uptake of dsRNA in *Ae. aegypti* was previously attempted using different sugar solutions, which differ in distribution to diverticula and foregut, but no uptake past the gut lumen was seen [33]. The uptake witnessed in *Cx. pipiens* tissues could potentially have occurred following the rupture of tissue during dissection. While possible, this is not likely to be the case here considering that dsRNA was also found in the head and legs, which were removed prior to the removal of the alimentary tract during dissections.

Following exposure, it is possible that fluorescent signal could be obfuscated by rapid degradation by nucleases or processing by Dicer-2. In *Ae. aegypti* adult females, Dicer-2 (AAEL006794) is expressed throughout mosquito tissues before and after bloodfeeding [59]. Other dsRNA nucleases are also known in mosquitoes, such as AAEL008858 and AAEL004103, which are expressed in larval alimentary tracts [20]. In locusts, dsRNA nucleases are expressed in the hemolymph, which possibly limits dsRNA uptake [50,60]; however, more work is needed to determine whether such nucleases are expressed in mosquitoes. While the presence of siRNAs or other small RNAs was not elucidated in this study, the fluorescent label chemistry used adds ~1 fluorophore per 10–20 base pairs of dsRNA, so fluorescent signal could be weakened through the production of siRNAs, but would not likely be entirely eliminated. As such, low levels of the signal may also be indicative of processed dsRNA.

Tracking dsRNA integrity by Northern blot revealed some full-length dsRNA to remain intact up to one week post exposure in adult *Ae. aegypti* (see Figure 2). A high dose of dsRNA was used in these experiments in order to detect signals following injection, but similar doses have been used before without inducing off-target knockdown effects [21]. Much of the dsRNA signal seen in mosquito tissues in the days following exposure appeared in the midgut lumen, indicating active excretion as has been witnessed before, but dsRNA was not seen in midgut cells. The processing or clearance of dsRNA was also apparently more rapid in midgut and abdominal tissues (containing the fat body). Both the midgut and the fat body are known to be susceptible to RNAi knockdown in a number of mosquito species. For instance, > 90% knockdown of the Target of Rapamycin gene has been achieved in both the fat body and midgut of *Ae. aegypti* [61,62]. As such, the lack of signal in the midgut and fat body may signify the rapid uptake and processing of dsRNAs. Albeit signal detection may have been blocked by a stronger signal from excreted dsRNA from the midgut lumen. The presence of dsRNA nucleases in the mosquito midgut means that dsRNAs could be rapidly degraded in the midgut lumen [20], but these nucleases are apparently differentially expressed in adult *Ae. aegypti* (see http://aegyptiatlas.buchonlab.com/ for AAEL008858 and AAEL004103, accessed 10 May 2023) [59]. DsRNA nucleases are also known to functionally differ between species [63]. As such, further work is needed to identify the degradation pathways of introduced dsRNAs, as well as the distribution of small RNAs generated from dsRNA.

Alternatively, the lack of signal could represent a lack of uptake, with the RNAi knockdown of target genes in the midgut and other tissues mediated by a secondary signal as part of the systemic RNAi response. In *Drosophila* and *Ae. Albopictus*, dsRNA derived from viruses can be reverse transcribed, re-expressed, and exported via exosomes [8,39]. The potential for a strong fluorescent signal to indicate the uptake and processing of dsRNA by the RNAi pathway in phagocytic cells is also supported. In mosquitoes and other Diptera, Dicer-2 can act to prevent the degradation of dsRNAs from defective viral genomes modulating immunity to viral infection [39]. As such, the prolonged presence of the fluorescent signal or Northern blot bands could indicate dsRNA bound by Dicer-2. Therefore, exogenously introduced dsRNAs could be sequestered by hemocytes and other phagocytic cells, and maintained for some time prior to siRNA generation. This would explain the longevity of knockdown witnessed in some mosquito studies, such as Chitin synthase silencing for 14 days in *Ae. aegypti* [64], AgKir1 potassium channel silencing for 11 days in *An. gambiae* [65], and superoxide dismutase silencing for 7 days in *Cx. pipiens* [66]. The longevity of dsRNA found among tissues in this study may indeed be linked to RNAi machinery binding to and protecting dsRNA from degradation by other nucleases [39]. This is evidenced by the rapid degradation of dsRNA found in the Dicer-2-deficient *Ae. albopictus* C6/36 cells compared to Dicer-2-competent mosquitoes in vivo (see Figure 2 and S18), which is known to lack an effective RNAi response, but rapidly produces short RNAs [40,67] reflective of an active PIWI pathway [68,69].

Beyond the protection of dsRNA, hemocytes are known to drive the spread of re-expressed RNAi triggers in *Drosophila* [19]. Our findings are in line with previous findings, as hemocytes are a major destination for dsRNA. It is interesting to hypothesize that a similar mechanism may occur with exogenously produced dsRNA in mosquitoes.

Our findings corroborate this mechanism, as a strong fluorescent iLacZ signal was present in hemocytes, and potentially re-expressed iLacZ bands were detected by Northern blots in tissues that lacked fluorescent signal. It is also possible that free iLacZ in the hemolymph, or hemocytes containing iLacZ, contaminated tissue samples when performing Northern blots, although all tissues were washed in PBS prior to RNA extraction.

Overall, this study provides an outline of dsRNA uptake and biodistribution following varying exposure routes in three medically significant mosquito species. We identified areas of dsRNA trafficking following uptake, and potentially super-sensitive tissues, such as the ovaries and hemocytes, where dsRNA accumulates. Finally, we addressed uptake and distribution to be relatively conserved among different species.

## 5. Conclusions

Overall, these results highlight that fluorescent RNAi triggers are taken up and sequestered by hemocytes and oocytes, and that this phenotype is conserved across mosquito species and life stages. It is possible that hemocytes play a role in spreading dsRNA, as they are a major site of dsRNA uptake following exposure. Beyond these cells, we saw evidence of a temporary association between iLacZ RNA and tissues, such as the Malpighian tubules and fat body, excreted through the alimentary tract following injection, indicating passage through alimentary tissues. Using Northern blots, we observed iLacZ RNA throughout the body with a dsRNA signal present up to one week post exposure.

## Figures and Tables

**Figure 1 insects-14-00556-f001:**
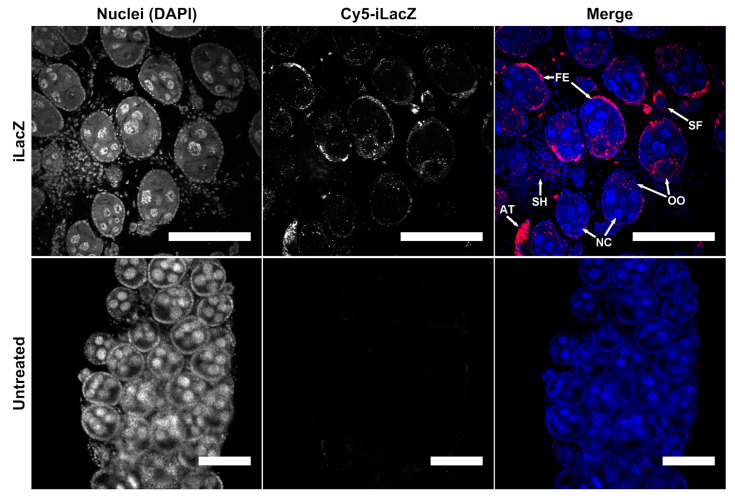
Multiple cell types uptake iLacZ in *Ae. aegypti* ovaries. Confocal microscopy of ovaries dissected 24 h post intrathoracic injection of iLacZ vs. no treatment control. Cell types shown include primary follicles composed of: follicular epithelia (FE), nurse cells (NC), and oocytes (OO), as well as secondary follicles (SF), epithelial cells of the ovarian sheath (SH), and atretic follicles (AT). Scale bar = 100 µm.

**Figure 2 insects-14-00556-f002:**
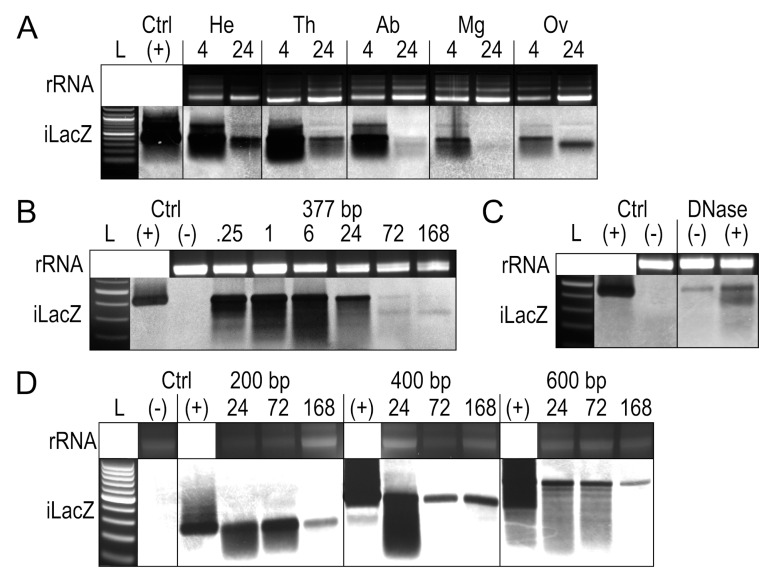
Tracking iLacZ by Northern blot in *Ae. aegypti* adult females. Overlaid gel electrophoresis and Northern blot runs following peritoneal exposure of iLacZ. (**A**) iLacZ in tissue extracts including the head (HE), thorax (TH), abdomen (AB), midgut (MG), and ovary (OV) at 4 and 24 h post exposure. (**B**) Time-course of iLacZ clearance in whole-body RNA extracts from 0.25 to 168 h post exposure. (**C**) Whole body extracts with iLacZ at 168 h post exposure with (+) and without (-) DNase treatment, single replicate performed. (**D**) Degradation time-course of a 200 bp, 400 bp, and 600 bp iLacZ products at 24, 72, and 168 h post exposure. L = 100 bp ladder, rRNA = ribosomal RNA loading controls imaged immediately prior to Northern membrane transfer, iLacZ = iLacZ 377 bp Northern blot probe, (+) ctrl = iLacZ stock solution, (-) untreated *Ae. aegypti* whole body RNA.

**Table 1 insects-14-00556-t001:** Tissue distribution of iLacZ signal in live *Ae. aegypti*, *An. gambiae*, and *Cx. pipiens*.

Species	HPE	Related Figures	Tissue-Affiliated Hemocytes						Intracellular Signal Locations					Extracellular Signal Locations						
			HD	LE	TH	HC-FB	TA-MT	TA-OV	PC	FB	VC	FO	AT	DV-CA	FG	MG	HG	RC	CT	IS
Peritoneal exposure by intrathoracic injection—adult female																				
*Ae. aegypti*	24–120	S1	+++	+++	++	+++	+++	++	++++	-	++	+++	-	-	-	-	++	++	-	++
*An. gambiae*	24–120	S2	++	-	+	++	+++	+++	++++	-	-	+++	-	-	-	+	+	+	-	++
*Cx. pipiens*	24–120	S3	+	*nd*	-	++	++	+++	++++	-	-	+++	-	-	+	+	++	+	-	+++
Peritoneal exposure by intrathoracic injection 24 h post bloodmeal—adult female																				
*Ae. aegypti*	1–24	S4	-	-	+	+	-	+	++++	-	-	+++	-	-	-	-	-	-	-	-
*An. gambiae*	1–24	S5	+++	++	-	+++	+	++	++++	-	-	+++	-	-	-	-	-	-	-	+
*Cx. pipiens*	1–24	S6	+	+	+	+++	+	+	++++	+	-	+++	-	-	-	-	-	-	-	++
Peritoneal exposure by intrathoracic injection—4th instar larvae																				
*Ae. aegypti*	24	S7	-	*na*	-	-	-	*na*	++	-	-	*na*	-	-	-	+	-	-	-	*na*
*An. gambiae*	24	S8	+	*na*	-	+	-	*na*	++	-	-	*na*	-	-	-	+	-	-	-	*na*
*Cx. pipiens*	24	S9	-	*na*	-	+	-	*na*	++	-	-	*na*	-	-	-	+	-	-	-	*na*
Topical exposure—adult female																				
*Ae. aegypti*	72	S10	-	-	-	-	-	-	-	-	-	-	-	-	-	-	-	-	++++	*na*
*An. gambiae*	72	S11	-	-	-	-	-	-	-	-	-	-	-	-	-	-	-	-	++++	*na*
*Cx. pipiens*	72	S12	-	-	-	-	-	-	-	-	-	-	-	-	-	-	-	-	++++	*na*
*Per os* exposure—adult female																				
*Ae. aegypti*	24–120	S13	-	-	-	-	-	-	+	-	-	-	-	+++	+++	+++	+++	+++	-	*na*
*Cx. pipiens*	24–120	S14	++	++	++	-	+	++	++	-	-	+	++	+++	+++	+++	+++	+++	-	*na*
Soaking exposure—1st instar larvae																				
*Ae. aegypti*	24–72	S15	*na*	*na*	-	-	-	*na*	-	-	-	*na*	*na*	*na*	*-*	+++	+++	+++	-	*na*

Symbols indicate: (-) no fluorescence found above background levels, (+) present in single replicate, (++) present on multiple occasions, (+++) present in every replicate, and (++++) present in every individual in every replicate. Tissue labels include: (HD) head, (LE) leg, (TH) thorax, (HC-FB) hemocoel and fat body, (TA-MT) tracheoles of Malpighian tubules, (TA-OV) tracheoles of ovary, (PC) pericardial cells, (VC) ventral nerve cord, (FO) follicles of ovary, (DV-CA) diverticula in adults-caecum in larvae, (FG) foregut lumen, (MG) midgut lumen, (HG) hindgut lumen, (RC) rectum lumen, (CT) cuticle, (IS) injection site. hours post exposure = Hours Post-Exposure, na = not applicable, nd = not sufficient data.

## Data Availability

All original data where not otherwise already provided are available on request from the corresponding author.

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
