# Peer review of "A Comparative Analysis of RNAi Trigger Uptake and Distribution in Mosquito Vectors of Disease"

_insects, 2023, doi:10.3390/insects14060556_

Round 1
Reviewer 1 Report
In this manuscript, Airs et al. compared the exogenous dsRNA taken by different tissues of Aedes aegypti, Anopheles gambiae, and Culex pipiens, and found that hemocytes and oocytes were the major tissues taken up and sequestered dsRNA as compared to other tissues. The manuscript could provide information about dsRNA experiment design regarding to gene functional study by dsRNA-mediated silencing of three mosquito species. Here are my comments.
1) In the siRNA-deficient mosquito cells (C6/36), the authors observed less iLacZ signal. I am confused about the results. Since siRNA is responsible for the degradation of dsRNA in insects and dicer2 is the enzyme to cut dsRNAs into 21-nt small RNAs, dsRNAs should be more stable in siRNA-deficient mosquito cells. Assuming that iLacZ RNA would be stable in C6/36 cells. But most iLacZ RNA seems to be degraded at 1 hours post exposure. It would be better if the authors could show whether iLacZ RNA was degraded by a trace of siRNA machinery in C6/36 cells or other nucleases (what nucleases?) as the authors stated.
2) If dsRNAs were taken by mosquito cells, it should be processed quicky by siRNA machinery if those cells had the entire siRNA pathway. iLacZ RNA was persistent for 24 hours post exposure in head, thorax, and ovaries, while iLacZ RNA was quickly degraded in the abdomen and midguts. The authors need to explain the dsRNA degradation in different tissues. Dose it relate to different siRNA machinery in tissues?
Line 359: it should be Figure 2C. It also needs to add the number of replicates for each northern blot image.
Author Response
Dear Editor and Reviewers. Firstly we would like to thank you for your time reviewing our manuscript and giving insightful comments and points which we address below. All comments are in regular text with author responses in bold type.
Reviewer 1 response
In this manuscript, Airs et al. compared the exogenous dsRNA taken by different tissues of Aedes aegypti, Anopheles gambiae, and Culex pipiens, and found that hemocytes and oocytes were the major tissues taken up and sequestered dsRNA as compared to other tissues. The manuscript could provide information about dsRNA experiment design regarding to gene functional study by dsRNA-mediated silencing of three mosquito species. Here are my comments.
1) In the siRNA-deficient mosquito cells (C6/36), the authors observed less iLacZ signal. I am confused about the results. Since siRNA is responsible for the degradation of dsRNA in insects and dicer2 is the enzyme to cut dsRNAs into 21-nt small RNAs, dsRNAs should be more stable in siRNA-deficient mosquito cells. Assuming that iLacZ RNA would be stable in C6/36 cells. But most iLacZ RNA seems to be degraded at 1 hours post exposure. It would be better if the authors could show whether iLacZ RNA was degraded by a trace of siRNA machinery in C6/36 cells or other nucleases (what nucleases?) as the authors stated.
Thank you for this comment. We have updated the discussion to reflect the points made below.
While C6/36 cells lack Dicer-2, they still can degrade dsRNAs as previously demonstrated in the literature. See C6/36 Aedes albopictus Cells Have a Dysfunctional Antiviral RNA Interference Response - PMC - “However, in C6/36 cells, viRNAs were primarily 17 nt in length from WNV infected cells and 26–27 nt in length in SINV and LACV infected cells.”
As such, dsRNA is known to degrade in these cells by other means, most probably in this case by the PIWI pathway see (https://pubmed.ncbi.nlm.nih.gov/26068474/ & https://pubmed.ncbi.nlm.nih.gov/22292064/), but defining these enzymes and the fate of dsRNA in this cell type was outside the scope and capacity of this work. This is why we only show this as a reference to in vivo work as a supplemental figure. If this experiment is insufficient we can remove it from the study and focus solely on in vivo findings, but thought this was an interesting demonstration of non-Dicer-2 degradation of dsRNAs.
2) If dsRNAs were taken by mosquito cells, it should be processed quicky by siRNA machinery if those cells had the entire siRNA pathway. iLacZ RNA was persistent for 24 hours post exposure in head, thorax, and ovaries, while iLacZ RNA was quickly degraded in the abdomen and midguts. The authors need to explain the dsRNA degradation in different tissues. Does it relate to different siRNA machinery in tissues?
The authors thank you for this comment. We have updated the text to discuss the potential for why the dsRNA might remain for longer in some tissues and consider the possibility of degradation by other nucleases. We noted in the discussion that Dicer-2 deficient C6/36 cells degrade dsRNA rapidly compared to in vivo. We have also added to the discussion examples of when knockdown of target genes is present for extensive periods, suggesting that dsRNA is possibly not degraded rapidly but processed over an extended period. While not peer reviewed, we have also compiled examples of knockdown across a wide range of RNAi studies https://airs.shinyapps.io/rnaidb/ and have included some examples from this analysis in the discussion where useful.
While we were unable to determine what nucleases act on dsRNAs in different tissues in mosquitoes, this study should lay the foundation for further work into determining different pathways for producing specific small RNAs or degraded products at the cell and tissue level.
Line 359: it should be Figure 2C. It also needs to add the number of replicates for each northern blot image.
Thank you for this comment, changes have been made to the text to this effect.
Reviewer 2 Report
RNA interference (RNAi) is a widely used tool in functional studies of mosquitoes and is being explored as a potential strategy for preventing infectious diseases transmitted by mosquito vectors. However, the effectiveness of RNAi-mediated gene knockdown varies among different target genes, tissues, and developmental stages. In this study, Airs et al. labeled a long dsRNA to a heterologous gene, LacZ, and conducted in vivo tracking experiments to monitor the distribution and stability of the dsRNA in three mosquito species. While the study provides valuable insights into how dsRNA spreads among mosquitoes, there are some issues that need to be addressed to maximize its significance and provide useful data for future experimental designs aimed at improving RNAi effectiveness in studies of mosquito vectors.
Major points:
1. The study did not clarify whether Dicer-2 can produce functional small interfering RNAs (siRNAs) from the Cy3-labeled dsRNAs used in the experiment, or whether these siRNAs can be detected by fluorescence microscopy. The answers to these questions could significantly affect the interpretation of the observed data.
2. The study did not provide clear information on the sample sizes used to report on the iLacZ signal distribution in live mosquitoes. Therefore, it is unclear whether the observed distributions are representative of all mosquitoes in the same sample group.
3. The Northern Blot analysis focused only on long dsRNA and ignored the mature siRNAs, which directly cause mRNA degradation and translational repression. Analyzing the iLacZ-derived siRNAs would provide useful insights into the diverse RNAi efficiencies in different tissues.
Author Response
Dear Editor and Reviewers. Firstly we would like to thank you for your time reviewing our manuscript and giving insightful comments and points which we address below. All comments are in regular text with author responses in bold type.
Reviewer 2
RNA interference (RNAi) is a widely used tool in functional studies of mosquitoes and is being explored as a potential strategy for preventing infectious diseases transmitted by mosquito vectors. However, the effectiveness of RNAi-mediated gene knockdown varies among different target genes, tissues, and developmental stages. In this study, Airs et al. labeled a long dsRNA to a heterologous gene, LacZ, and conducted in vivo tracking experiments to monitor the distribution and stability of the dsRNA in three mosquito species. While the study provides valuable insights into how dsRNA spreads among mosquitoes, there are some issues that need to be addressed to maximize its significance and provide useful data for future experimental designs aimed at improving RNAi effectiveness in studies of mosquito vectors.
Major points:
- The study did not clarify whether Dicer-2 can produce functional small interfering RNAs (siRNAs) from the Cy3-labelled dsRNAs used in the experiment, or whether these siRNAs can be detected by fluorescence microscopy. The answers to these questions could significantly affect the interpretation of the observed data.
Thank you for this point. While we do not determine whether Dicer-2 processes the labeled dsRNA, but this kit has been demonstrated to not impair Dicer-2 activity in insects previously (see: https://doi.org/10.1016/j.jinsphys.2013.01.009) and has been cited in the text to direct readers to prior success of this method. We have also performed in house tests and find knockdown is not inhibited by the Mirus kit, see the figure attached above (to reviewer 1 comments) as evidence of this.
The kit conjugates a fluorophore to every 10 bases, and we have added a note about this in the discussion relating to potential for siRNAs / processed dsRNA to yield a weaker or undetectable staining profile compared to full length dsRNA. Product literature is linked below: https://www.mirusbio.com/products/labeling/label-it-nucleic-acid-labeling-reagents#product:MIR%203625
- The study did not provide clear information on the sample sizes used to report on the iLacZ signal distribution in live mosquitoes. Therefore, it is unclear whether the observed distributions are representative of all mosquitoes in the same sample group.
Thanks for spotting this! We have added the following details to the text. “At each timepoint for adult stage exposure groups of 10 individuals were selected at random and dissected for analyses.” and “For larval analyses groups of 5 were selected per timepoint.”
- The Northern Blot analysis focused only on long dsRNA and ignored the mature siRNAs, which directly cause mRNA degradation and translational repression. Analyzing the iLacZ-derived siRNAs would provide useful insights into the diverse RNAi efficiencies in different tissues.
Thank you for this comment. Because a number of different species, life stages, and exposure routes were explored in this paper, we did not have the capacity to also perform in depth analyses of siRNA generation which would require small RNA sequencing to truly resolve the small RNA product generation from iLacZ dsRNAs.
We had hoped to perform small RNA sequencing following injection of dsRNAs, but lacked the budget and time to perform these studies in this work. To work around this limitation, we opted to visualize dsRNA degradation using Northern blotting, but do certainly acknowledge that siRNAs are difficult to resolve this way because they may have run off the gel, or have minimum affinity to the ssRNA probe. Ideally profiling of siRNAs is the next step and text has been added to the discussion to this point.
Round 2
Reviewer 2 Report
The authors have adequately addressed the concerns raised in the review.
Author Response
Thank you for your time and insightful comments in reviewing this manuscript.